# GUY1 confers complete female lethality and is a strong candidate for a male-determining factor in *Anopheles stephensi*

Frank Criscione[†], Yumin Qi, Zhijian Tu*

Department of Biochemistry, Virginia Tech, Blacksburg, United States

**Abstract** Despite their importance in sexual differentiation and reproduction, Y chromosome genes are rarely described because they reside in repeat-rich regions that are difficult to study. Here, we show that *Guy1*, a unique Y chromosome gene of a major urban malaria mosquito *Anopheles stephensi*, confers 100% female lethality when placed on the autosomes. We show that the small GUY1 protein (56 amino acids in length) causes female lethality and that males carrying the transgene are reproductively more competitive than their non-transgenic siblings under laboratory conditions. The GUY1 protein is a primary signal from the Y chromosome that affects embryonic development in a sex-specific manner. Our results have demonstrated, for the first time in mosquitoes, the feasibility of stable transgenic manipulation of sex ratios using an endogenous gene from the male-determining chromosome. These results provide insights into the elusive M factor and suggest exciting opportunities to reduce mosquito populations and disease transmission.

*For correspondence: jaketu@vt.edu

Present address: [†]Entomology, Institute for Bioscience and Biotechnology Research, University of Maryland, Rockville, United States

Competing interests: The authors declare that no competing interests exist.

## Introduction

Insects employ diverse sex-determination mechanisms at the chromosomal level including XX/XY, ZW/ZZ, XX/XO, and diploid/haploid chromosomal systems (*Bopp et al., 2014*; *Bachtrog et al., 2014*; *Biedler and Tu, 2016*). Similarly, the primary molecular signals that determine sex are highly divergent in insects. In honeybees which use the diploid/haploid system, heterozygosity of the *complementary sex determiner (csd)* gene is the primary signal that initiates female development (*Hasselmann et al., 2008*). In silkworms a piRNA gene on the W chromosome triggers female development (*Kiuchi et al., 2014*). In the fruit fly *Drosophila melanogaster*, which has XX/XY sex chromosomes, the collective dose of X-linked signal elements (XSE) functions as the signal that specifies sex in the early embryo (*Erickson and Quintero, 2007*). However, a dominant male-determining factor (*M*) on the Y chromosome initiates male development in many other insects that contain XX/XY sex chromosomes (*Sanchez, 2008*; *Biedler and Tu, 2016*). Instead of a well differentiated and heteromorphic Y chromosome, mosquitoes of the *Culicinae* subfamily, which includes the genus *Aedes*, contain a homomorphic sex-determining chromosome that harbors an *M* factor in the male-determining locus (*M*-locus). A novel RNA-binding protein named *Nix* was recently shown to be an *M* factor in *Aedes aegypti* (*Hall et al., 2015*).

Despite rapid changes in the primary signals or master switches, two key transcription factors at the bottom of the sex-determination pathway, *doublesex (dsx)* and *fruitless (fru)*, are highly conserved in insects. Sex-specific splicing of *dsx* and *fru* pre-mRNAs leads to the production of sex-specific DSX and FRU protein isoforms, which program sexual differentiation (*Bopp et al., 2014*). The alternative splicing of *dsx* and *fru* pre-mRNAs is often controlled by a protein complex that includes

**eLife digest** Much like humans, *Anopheles* mosquitoes have a pair of sex chromosomes that determine whether they are male or female: females have two X chromosomes, while males have an X and a Y. Genetic evidence has indicated that there is a dominant male-determining factor on the Y chromosome that acts as a master switch to cause mosquitoes to develop into males. Mosquitoes that lack a Y chromosome, and hence the male-determining factor, therefore develop into the default female sex.

Because only female mosquitoes feed on blood and transmit disease-causing microbes – including those that cause malaria – there is strong interest in identifying the male-determining factor. Introducing this gene into females could allow mosquito sex ratios to be manipulated towards the harmless non-biting males.

In 2013, a study of male *Anopheles stephensi* mosquitoes identified a gene called *Guy1* that is only found on the Y chromosome. Criscione et al. – who were involved in the 2013 study – now show that female *A. stephensi* mosquitoes die when the *Guy1* gene is placed on their non-sex chromosomes.

Further investigation confirmed that the protein produced from the *Guy1* gene kills the females. This protein is an initiating signal that affects embryonic development in a sex-specific manner, making it a strong candidate to be the male determining factor in *A. stephensi.* This is consistent with previous reports in which the master switches of sex determination could be manipulated to kill specific sexes in fruit flies and nematode worms.

Criscione et al. also found that males that carry the inserted *Guy1* gene on their non-sex chromosomes – and so could potentially pass it on to both male and female offspring – are reproductively more competitive than their non-modified siblings under laboratory conditions. As the resulting female offspring would not survive, it is thus feasible, in principle, to genetically manipulate the sex ratio of the mosquitoes.

A future challenge will be to identify how the protein encoded by the *Guy1* gene acts to kill female mosquitoes. This knowledge will help to investigate the feasibility of using genetically modified mosquitoes to reduce *Anopheles* populations in order to control malaria.

a fast-evolving *transformer* (TRA) and a conserved *transformer 2* (TRA2), where TRA is the sex-specific protein in this TRA/TRA2 complex. Indeed, TRA often functions as an intermediate that transduces the selected sexual fate from the primary signal to the DSX and FRU effector molecules (*Bopp et al., 2014*). For example, in the medfly *Ceratitis capitata*, which has XX/XY sex chromosomes, a functional TRA is produced in the zygotic XX embryo as a result of splicing by a maternally deposited TRA/TRA2 complex, leading to female-specific *dsx* and *fru* splicing and thus the female sex, which is then maintained by the self-sustaining loop of *tra* splicing and function. In males, a yet-to-be-discovered M factor interrupts this loop of *tra* splicing, leading to the male sex (*Pane et al., 2002*). Sex-specific splicing of *dsx* and *fru* has been described in both *Aedes* and *Anopheles* mosquitoes, suggesting the presence of a TRA-like activity (*Scali et al., 2005*; *Salvemini et al., 2011*, *2013*). However, a *tra* gene or its functional homolog has not yet been found in any mosquitoes.

*Anopheles* mosquitoes contain well-differentiated X and Y chromosomes and genetic evidence suggests that a dominant *M factor* on the Y chromosome controls male development in *Anopheles* mosquitoes (*Baker and Sakai, 1979*). The *Anopheles* Y chromosome also regulates mating behavior (*Fraccaro et al., 1977*). There is tremendous interest in deciphering Y gene function in non-model insects, including *Anopheles* mosquitoes, to shed light on the mechanism and evolution of sexual differentiation. There has also been strong interest in Y chromosome genes in *Anopheles* mosquitoes for translational motivations. Only female mosquitoes transmit disease pathogens because only females feed on blood. Thus, it is ideal, if not required, to release only males when considering genetic approaches for reducing mosquito populations or for replacing competent vector populations with populations that are refractory to disease transmission (e.g., [*Collins, 1994*; *Benedict and Robinson, 2003*; *Windbichler et al., 2008*; *Fu et al., 2010*; *Black et al., 2011*; *Harris et al., 2012*]). A better understanding of the Y chromosome function in sex-determination and male reproduction

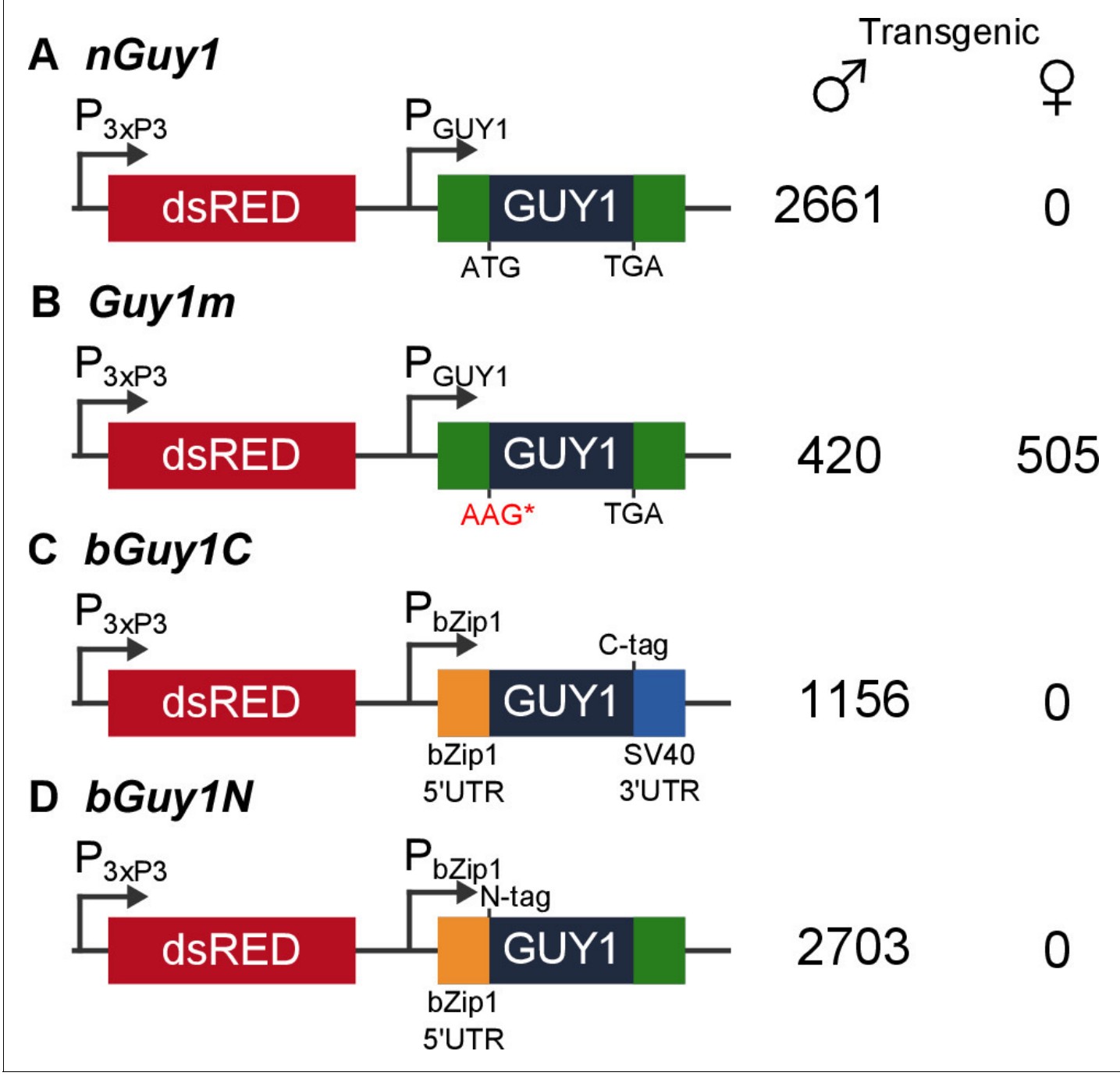

**Figure 1.** Four donor plasmids were used to generate transgenic *Anopheles stephensi*. *nGuy1* and *Guy1m* were also used in the transient assays described in *Table 1*. All constructs shown in the figure were flanked by the *piggyBac* arms to facilitate *piggyBac*-mediated integration into the *An. stephensi* genome (*Horn et al., 2000*). The DsRed fluorescent marker gene under the control of the 3xP3/Hsp70 promoter (3xP3) was the transformation marker. $P_{GUY1}$ refers to the native Guy1 promoter (*Criscione et al., 2013*). Note that the only difference between *nGuy1* and *Guy1m* is the point mutation in the first ATG. $P_{bZip1}$ refers to a promoter derived from an *An. stephensi* gene (Genbank JQ266223) and this promoter is used to drive early zygotic expression of the transgene (*Figure 3*). The C-tag and N-tag refer to the eight residue Strep II tag (*Lichty et al., 2005*) placed at either the C- or N-terminus of the GUY1 protein. A stretch of eight glycine residues were placed between the Strep II tag and the GUY1 protein (*Supplementary file 3*). The number of transgenic males and females were total counts from screens performed on all lines of each construct (*Supplementary files 1* and *2*).

**Table 1.** Transient injection of the *nGuy1,* but not the *Guy1m,* plasmid in early embryos confers strong male bias in *Anopheles stephensi*\*.

| Plasmid | Male | Female |
| --- | --- | --- |
| *nGuy1,* replicate 1 | 25 | 1 |
| *nGuy1,* replicate 2 | 19 | 2 |
| *Guy1m,* replicate 1 | 24 | 34 |
| *Guy1m,* replicate 2 | 10 | 6 |

Notes:

\*An EGFP reporter plasmid under the control of the *Drosophila melanogaster* actin 5C promoter was co-injected with either the *nGuy1* or *Guy1m* plasmid to ensure effective embryonic injections as indicated by an EGFP signal in the larvae. The adults that developed from EGFP positive larvae were sexed according to antennae morphology.

may provide novel targets for interference and lead to several practical applications. For example, transgenic lines may be obtained that produce male-only mosquitoes, resulting in more cost-effective mass production and sex separation methods than current approaches (e.g., [*Papathanos et al., 2009*]). Furthermore, releasing such transgenic males is theoretically much more efficient than classic sterile insect techniques in achieving population reduction and disease control because of the added benefit of male-bias in subsequent generations (*Thomas et al., 2000*; *Schliekelman et al., 2005*).

Despite strong interest and the availability of genomic resources (e.g., [*Holt et al., 2002*]), earlier systematic efforts failed to identify Y genes in *Anopheles* mosquitoes (*Krzywinski et al., 2006*). Several Y genes were recently discovered in *An. stephensi* and *An. gambiae* (*Criscione et al., 2013*; *Hall et al., 2013*) by sequencing males and females separately. None of these genes is homologous to the *Nix* gene in *Ae. aegypti* (*Hall et al., 2015*) or encodes a predicted RNA-binding protein or splicing factor. However, among the *An. gambiae* Y chromosome genes, *gYG2* (*An. gambiae* Y Gene 2) is the most likely candidate for the *M* factor because it showed early embryonic expression (*Hall et al., 2013*) and it is the only Y gene that is shared among all species within the *An. gambiae* species complex (*Hall et al., 2016*). In a recent report, *gYG2* (renamed YOB by the authors) was shown to confer female-specific lethality in a transient embryonic assay and shift *doublesex (dsx)* splicing towards the male isoform in an *An. gambiae* cell line, suggesting that gYG2/YOB functions as an *M* factor in *An. gambiae* (*Krzywinska et al., 2016*). Among the four Y genes identified in *An. stephensi*, *Guy1* is the best candidate for the *M factor* because it is transcribed the earliest among all Y genes, at the very onset of embryonic development (*Criscione et al., 2013*).

## Results

### Transient embryonic injection of *Guy1* caused male bias

*Guy1* transcription is transient and its transcript tapers off approximately 8–12 hr after egg deposition (*Criscione et al., 2013*). We designed a plasmid, *nGuy1* (*Figure 1A*), which contains the entire *Guy1* gene with its native promoter that was shown to function in the early embryo (*Criscione et al., 2013*), to first test the function of *Guy1* in *An. stephensi* by a transient embryonic assay. A plasmid that contains an enhanced green fluorescent protein (EGFP) marker controlled by *D. melanogaster* actin 5C promoter was co-injected to select for effective embryonic injections as indicated by the EGFP signal in the larvae. The adults that developed from EGFP positive larvae showed 25:1 and 19:2 male to female ratio (*Table 1*). When the same experiments were performed using *Guy1m* (*Figure 1B*), a plasmid that is identical to *nGuy1* except for a point mutation that changed the start codon ATG to AAG, the male-bias phenotype was no longer observed (*Table 1*), suggesting that the GUY1 protein is the cause of the phenotype.

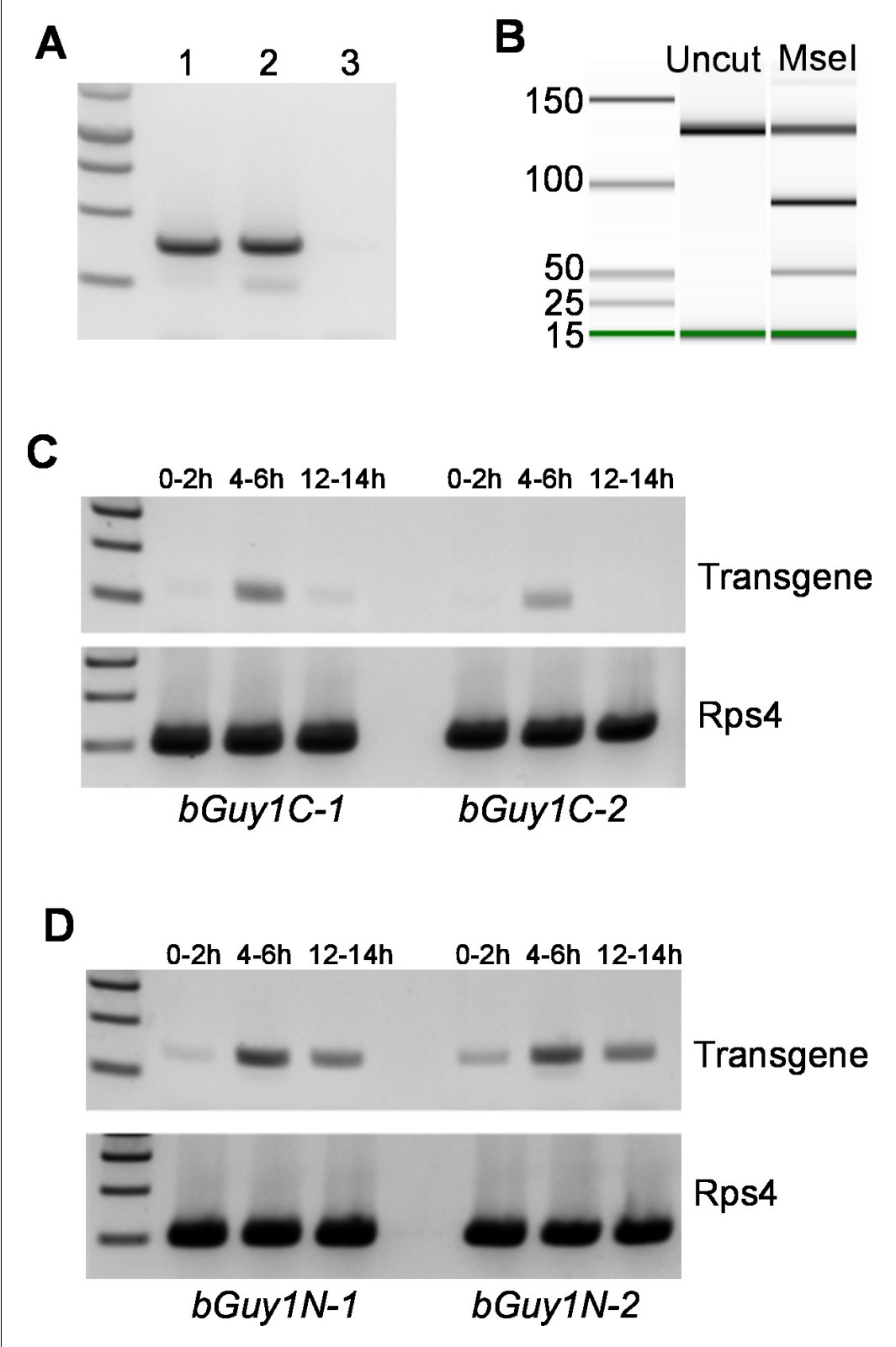

**Figure 2.** Early embryonic expression of the *Guy1* transgene in *nGuy1* (**A**), *Guy1m* (**B**), *bGuy1C* (**C**), and *bGuy1N* (**D**) lines. (**A**) RT-PCR from 5–6 hr old genotyped single embryos. Lane 1, Wild type male; Lane 2, Transgenic female; Lane 3, Wild type female. Genotyping was performed according to *Criscione et al. (2013)*. *nGuy1* transcription is detected in transgenic females where there is no Y chromosome or endogenous *Guy1*. Detecting *nGuy1* expression in transgenic female embryos is the only way we can show that the *nGuy1* transgene is expressed because there is no sequence difference

*Figure 2 continued on next page*

*Figure 2 continued*

between the *nGuy1* transgene transcript and the endogenous *Guy1* transcript. (B) A pseudo-gel image from the bioanalyzer (Agilent Technologies, Inc., Santa Clara, CA, USA) showing the presence of RT-PCR products from the mutant *Guy1m* transgene. We took advantage of the *Guy1m* mutation (ATG to AAG) that introduced an MseI site so we do not have to perform RT-PCR on genotyped single embryos. RT-PCR was done using pooled 5–6 hr embryos. The three fragments in the MseI lane are 130 bp (uncut wildtype *Guy1*), 90 bp (cut *Guy1m*), and 40 bp (cut *Guy1m*). The presence of the MseI digested 90 and 40 bp fragments indicates that there were transcripts from the *Guy1m* transgene. Panels **C** and **D** show RT-PCR products using a *Guy1* primer and a *bZip1* UTR primer, which only amplify cDNA from the *bGuy1C* and *bGuy1N* transgenes (**Figure 1**). Eggs of 0–2 hr, 4–6 hr and 12–14 hr post oviposition were collected from *A. stephensi bGuy1C-1* and *bGuy1C-2* (panel **C**) and *bGuy1D-1* and *bGuy1D-2* (panel **D**) lines. RPS4, ribosomal protein subunit 4. All primers can be found in **Table 3**.

## Transgenic lines that ectopically express the native *Guy1* gene produced only male transformants over 15 generations

Both *nGuy1* and *Guy1m* are *piggyBac* donor plasmids that contain the DsRed transformation marker (**Figure 1**). We were able to obtain transgenic lines using these donor plasmids together with a *piggyBac* helper plasmid to achieve germ line transposition. Two independent *nGuy1* transgenic lines and one *Guy1m* line were obtained. Only a single transgene was inserted in each line, as indicated by digital droplet PCR (not shown), a method that quantifies gene copy number (**Hindson et al., 2011**). Inverse PCR and sequencing revealed independent insertion of the transgene in the two *nGuy1* lines as the two *nGuy1* insertion sites mapped to two different scaffolds (**Supplementary file 3**). Not a single transgenic female older than the third instar has been detected since we established the *nGuy1* lines, while 2661 transgenic positive males were identified over 15 generations (**Figure 1** and **Supplementary file 1**). Similar to what was observed during the transient assay, the *Guy1m* line showed no male-bias (**Figure 1B** and **Supplementary file 1**). Both *nGuy1* and *Guy1m* transgenes are expressed in the early embryos in their respective lines (**Figure 2A and B**). **Table 2** shows the numbers of transgene positive and negative males and females of the seventh generation (G$_7$). The lack of sex-bias in the non-transgenic progeny in all three lines indicates autosomal insertion of the transgenes.

## Further evidence suggests that the 56 amino acid GUY1 protein is responsible for the lack of female transformants

Two additional transgenic lines were obtained using *bGuy1C* (**Figure 1C**), which contains the 168 bp *Guy1* open reading frame (ORF) plus a C-terminal Strep II tag (**Lichty et al., 2005**). The expression

**Table 2.** Sex ratios of transgenic and non-transgenic progeny of five transgenic lines*.

| Line | Transgenic male | Transgenic female | Negative male | Negative female |
|---|---|---|---|---|
| nGuy1-1 [†] | 71 | 0 | 58 | 56 |
| nGuy1-2 [†] | 129 | 0 | 91 | 83 |
| Guy1m [†] | 69 | 62 | 26 | 34 |
| bGuy1C-1 [‡] | 78 | 0 | 75 | 66 |
| bGuy1N-1 [‡] | 77 | 0 | 66 | 65 |

Notes:

*Crosses were done between the heterozygous transgenic males and wildtype females. Screening and sexing was initially done at the L3 instar stage. The sexed negative and positive larvae were reared separately to adulthood and sex was further confirmed on the basis of antennae morphology. The total numbers of each sex for the transgenic and non-transgenic groups are listed.

[†]The numbers for these lines are from generation 7 (G$_7$). Note that *Guy1m* has a point mutation that abolished the *Guy1* open reading frame. There are more transgenic individuals than non-transgenic individuals in the *Guy1m* line because both transgenic females and wild type females were mated with transgenic males to maintain the line.

[‡]Two *bGuy1C* lines and nine *bGuy1N* lines were obtained and only one of each is shown in this Table. The numbers are from generation 4 (G$_4$).

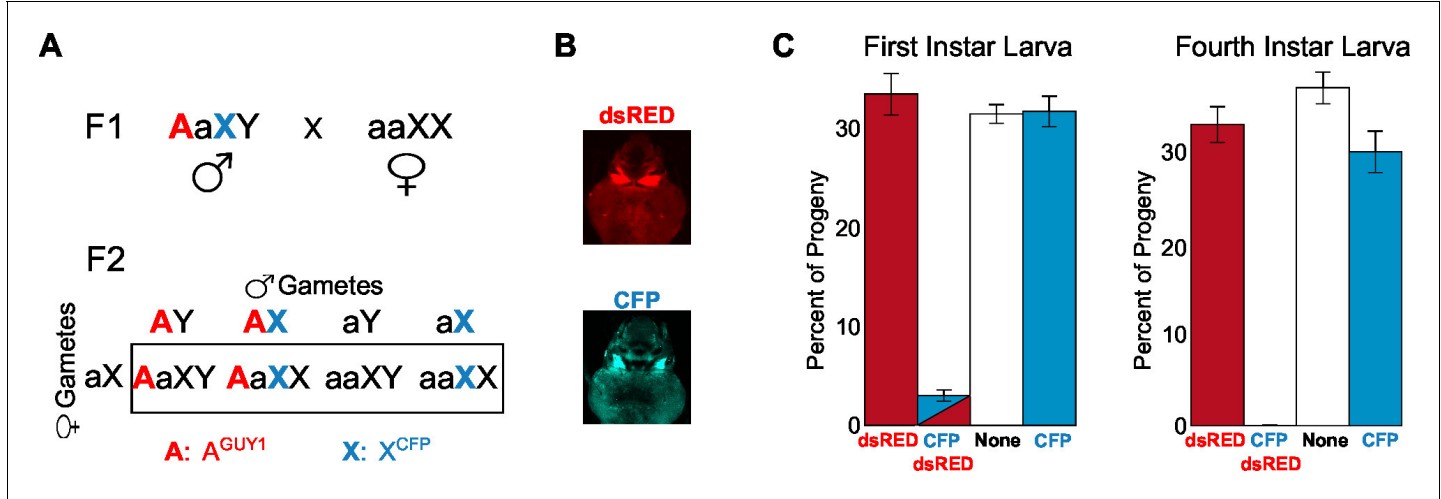

**Figure 3.** Analysis of the transgenic *nGuy1-1* line indicates 100% female lethality prior to or soon after egg hatching. (**A**) A schematic and Punnett square showing the cross performed to genotype F2 progeny based on expression of fluorescent transformation markers. The red uppercase A indicates an autosome that carries the Guy1 transgene and the dsRED transformation marker gene, or $A^{GUY1}$. The cyan uppercase X indicates an X chromosome that carries a Cyan fl protein (CFP) transformation marker gene, or $X^{CFP}$. $A^{GUY1}$aXY males and aa$X^{CFP}X^{CFP}$ females were initially crossed to obtain F1 $A^{GUY1}$a$X^{CFP}$Y progeny. $A^{GUY1}$a$X^{CFP}$Y were then crossed with wild type females (aaXX) to obtain progeny of four possible genotypes: $A^{GUY1}$aXY, transgenic *Guy1* males; $A^{GUY1}$a$X^{CFP}$X, transgenic *Guy1* females; aa$X^{CFP}$X, wild type females; aaXY, wild type males. (**B**) Images of transgenic L3 instar showing DsRed positive and CFP positive, respectively. (**C**) Distribution of the four genotypes in the F2 progeny at L1 and L4 instar stages, respectively. Analysis of the L1 and L4 instar is from two independent experiments. Percentages of each genotype were shown as the average of four replicates with standard error. The actual count of each genotype is provided in *Figure 3—source data 1*. Note that all CFP-DsRed double positive L1 instars died within 8 hr after hatching.

The following source data is available for figure 3:

**Source data 1.** Number of the four types of progeny from wild type males mated with *nGuy1-1* (DsRed) and CFP positive males.

of *bGuy1C* is controlled by an *An. stephensi* early zygotic promoter derived from the *bZip1* gene (Genbank JQ266223). The 5'UTR is provided from *bZip1* and the 3' UTR from SV40. Thus, the only *Guy1* sequence in *bGuy1C* is the ORF. Nine additional transgenic lines were obtained using *bGuy1N* (*Figure 1D*), a plasmid similar to *bGuy1C* except that the Strep II tag is at the N-terminus and the 119bp *Guy1* 3' UTR was used instead of the SV40 3' UTR. These 11 lines were produced from different $G_0$ pools and, thus, are likely all independent. The *bGuy1C* lines produced 1156 transgenic males and 0 transgenic females and all nine *bGuy1N* lines produced 2703 transgenic males and 0 transgenic females in $G_{2-4}$ (*Figure 1* and *Supplementary file 2*). The early zygotic transcription of the *Guy1* transgene is shown in *Figure 2C and D*. Transgenic lines that contain a functional *Guy1* gene (*nGuy1*, *bGuy1C*, and *bGuy1N*) produced 6520 transgenic males and zero transgenic females. These results further indicate that the GUY1 protein is the cause of the lack of females because the common feature of these lines is the *Guy1* ORF while the promoters, 5' and 3' untranslated regions differ.

## Female-killing, not sex conversion, is the cause of the male-only phenotype

The observation that all transgenic mosquitoes were males could result from either lethality or sex conversion of the XX individuals. We took advantage of an existing transgenic line that has a cyan fluorescent protein (CFP) marker cassette inserted on the X chromosome ($X^{CFP}$) (*Amenya et al., 2010*) to monitor the genotype. The presence of the *nGuy1* transgene on an autosome ($A^{GUY1}$) is monitored by the DsRed marker. *Figure 3A* shows the schematic of the crosses performed to produce progeny whose genotype can be monitored by CFP and DsRed expression (*Figure 3B*). First, $A^{GUY1}$aXY males were mated with aa$X^{CFP}X^{CFP}$ females to obtain the $A^{GUY1}$a$X^{CFP}$Y F1 progeny. These $A^{GUY1}$a$X^{CFP}$Y males were subsequently crossed with wild type females (aaXX) to obtain progeny with

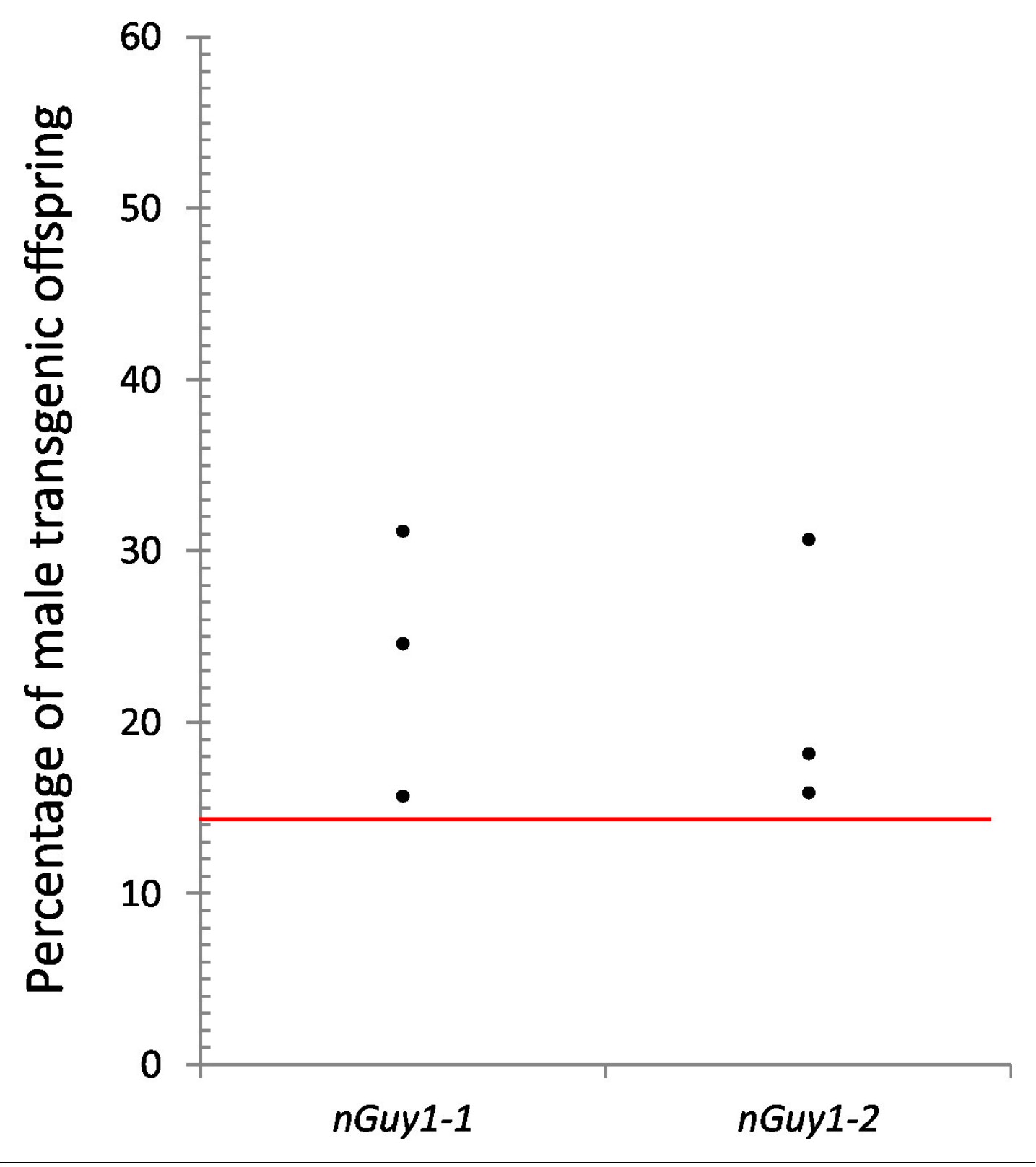

**Figure 4.** Reproductive competitiveness of transgenic males compared to their non-transgenic male siblings in two independent lines, *nGuy1-1* and *nGuy1-2*. Sibling cohorts of 20 transgenic and 20 non-transgenic males were mated with 10 wild type females. The resulting progeny were screened for transgenes at L3 instar stage as indicated by the DsRed marker and sexed by the presence or absence of testes. Shown in the figure percentages of male transgenics in biological triplicates for both lines. The red line indicates the expected percentage (1/7 or 14.29%) of transgenic male progeny,
*Figure 4 continued on next page*

*Figure 4 continued*

assuming that the DsRed positive females do not survive beyond the L1 stage (*Figure 3*) and the male parents (transgenics and their non-transgenic brothers) were equally productive (detailed calculations are shown in *Figure 4—source data 1*). Statistical analysis was performed using one-sample proportion tests for both lines (Z=5.0 and 8.1, respectively; p<0.001 in both cases, *Figure 4—source data 1*). The significantly larger transgenic male population in comparison to the expected value suggests that the transgenic *Guy1* containing males are reproductively more competitive than non-transgenic males under these laboratory conditions.

The following source data is available for figure 4:

**Source data 1.** Assay for male reproductive competitiveness of *nGuy1-1* and *nGuy1-2* lines [1].

four possible genotypes that can be differentiated based on CFP and DsRed expression: (1) $A^{GUY1}aXY$, DsRed positive and CFP negative; (2) $A^{GUY1}aX^{CFP}X$, DsRed positive and CFP positive; (3) $aaX^{CFP}X$, DsRed negative and CFP positive; and (4) $aaXY$, DsRed negative and CFP negative. As shown in *Figure 3C* and *Figure 3—source data 1*, there is a complete lack of DsRed/CFP double positives when the F2 progeny were screened at 4th instar larvae (L4), indicating the lack of $A^{GUY1}aX^{CFP}X$ larvae. All DsRed positives were confirmed to be males by the presence of a pair of testes. Sexing method was confirmed by allowing the larvae to develop into adults. We then repeated these experiments by screening the L1 instars to determine the timing of the death of $A^{GUY1}aX^{CFP}X$ individuals. In four replicates, only 2–4% of the L1 larvae showed the $A^{GUY1}aX^{CFP}X$ genotype and they all died within 8 hr of hatching. Thus, we have seen complete lethality of transgenic XX individuals prior to, or soon after, egg hatching.

## Reproductive competitiveness of the *nGuy1* males under laboratory conditions

*Figure 4* and *Figure 4—source data 1* show the results of the experiments to assess the reproductive competitiveness of the *nGuy1* males under laboratory conditions. To mitigate the effect of different genetic backgrounds, transgenic males and their non-transgenic sibling males were subjected to competition for wild type virgin females. Two days after emergence, 20 transgenic males and 20 of their non-transgenic male siblings were placed in a 44 oz cup to compete for 10 wild type virgin females. The progeny of these females was screened for the *Guy1* transgene and sexed at the L3 larval stage. Biological triplicates were performed for both *nGuy1* lines. It was experimentally determined that transgenic females do not survive to the L3 stage (*Figure 3C*). Thus, the expected fractions for transgenic males, non-transgenic females, and non-transgenic males were 1/7, 3/7, and 3/7, respectively, assuming equal reproductive capability of the *nGuy1* males and their siblings (see *Figure 4—source data 1* for genotype explanation). In all three experiments and for both lines, there are more transgenic *nGuy1* offspring than expected under the assumption of equal reproductive ability (*Figure 4* and *Figure 4—source data 1*). The results are statistically significant for both *nGuy1* transgenic lines according to one-sample proportion tests (Z=5.0 and 8.1, respectively; p<0.001 for both lines, *Figure 4—source data 1*). The significantly larger transgenic male population in comparison to the expected value suggests that the transgenic *nGuy1* males were more competitive in reproductive output than their non-transgenic male siblings in the laboratory under our assay conditions.

## *Guy1* expression does not require other factors from the Y chromosome

We have previously shown that *Guy1* is the earliest transcribed of all known Y genes in *An. stephensi*, at the very onset of maternal-to-zygotic transition (*Criscione et al., 2013*). Here, we have shown that the native *Guy1* promoter is active in the early embryos of both *nGuy1* and *Guy1m* lines (*Figure 2A and B*). Furthermore, the *Guy1* promoter is active in female embryos, according to direct evidence by RT-PCR (*Figure 2A*) and according to the fact that *nGuy1* transgene affects females during the embryonic stage. These results from transgenic lines clearly indicate that the early embryonic gene expression controlled by the native *Guy1* promoter does not rely on any other Y chromosome factors because Y chromosome is not present in the female embryos.

**Table 3.** Primers and probes used in this study.

| | |
|---|---|
| GUY1_Full-F1 | CCTGAAATGATGCTCTGGAAA |
| GUY1_Full-R1 | GAAACGTTTTTCCAACATGTGA |
| GUY1_Full+AscI-F1 | CACTGGCGCGCCCCTGAAATGATGCTCTGGAAA |
| GUY1_Full+PacI-R1 | TCATGCAATTAATTGAAACGTTTTTCCAACATGTGA |
| RPS4-F2 | GAGTCCATCAAAGGAGAAAGTCTAC |
| RPS4-R2 | TAGCTGGCGCATCAGGTAC |
| sYG2_F2 | TGCCGGACATGACATTTG |
| sYG2_R2 | TCAATGCGAACAGAAGGCTAA |
| DsRed_F2 | CCCCGTAATGCAGAAGAAGA |
| GUY1_R2 | GATCCGTTAAAAATTGACACCA |
| GUY1_F3 | TTTACTCGTCAAAGCTGCCA |
| GUY1_R3 | GATCCGTTAAAAATTGACACCA |
| DsRed_F4 | CCCCGTAATGCAGAAGAAGA |
| DsRed_R4 | GGTGATGTCCAGCTTGGAGT |
| DsRed_P4 | FAM–TACATGGCCAAGAAGCCCGT–BHQ1 |
| AutoRef_ddP_F4 | ATCACCACTCGTCGTCCGTT |
| AutoRef_ddP_R4 | CGAACGAACTCGATTGACCC |
| AutoRef_ddP_P4 | HEX–GCAAACACCACAACAGCAGC–BHQ1 |
| *GUY1*_ddP_F4 | GTCAAAGCTGCCACGGATCT |
| *GUY1*_ddP_R4 | TCCAATGTCACAGCAGAGTGTTT |
| *GUY1*_ddP_P4 | FAM–TCACAAAGTAGGCGATACAAAAACA–BHQ1 |
| iPCR_PBR_F5 | TACGCATGATTATCTTTAACGTA |
| iPCR_PBR_R5 | TGGCTCTTCAGTACTGTCAT |
| iPCR_PBRnest_F5 | GTCACAATATGATTATCTTTCTA |
| iPCR_PBRnest_R5 | CACTTCATTTGGCAAAATAT |
| Guy1m_F6 | TTAGATAACAGAAAGCGTACACT |
| Guy1m_R6 | ACTTGATTTATCATTCCAAGTCA |
| bGuy1C-F7 | GTTTATTGCAGCTTATAATGGTTAC |
| bGuy1C-R7 | TCTGACTTGGAATGATAAATCAAG |
| bGuy1N_F7 | GAGAAAATAGCTGAATTGAAGG |
| bGuy1N_R7 | AACACACAAAGTGGTCTTATGC |
| RPS4_F7 | CACGAGGATGGATGTTGGAC |
| RPS4_R7 | ATCAGGCGGAAGTATTCACC |

Notes: Primers are presented in seven sets which alternates in white and grey backgrounds. (1) Amplification of the full length *Guy1* gene sequence followed by adding a 5' adaptor with an AscI site and a 3' adaptor with a PacI site. (2) Primers used for genotyping embryos. RPS4, ribosomal protein subunit 4, was used as the positive control, *sYG2* was used as a Y chromosome marker, and the transgene was amplified using a DsRed and a *Guy1* primer, which is only present in transgenic individuals. (3) Primer sets used for RT-PCR of *Guy1* on single embryos. (4) Digital droplet PCR primer sets used to determine *Guy1* copy number and number of insertions in transgenic lines. (5) Inverse PCR primer sets used to determine the site of integration near the *piggyBac* right hand flanking sequences. (6) RT-PCR primers to amplify a 130 bp fragment of *Guy1* to allow differentiation of *Guy1m* transcript from endogenous *Guy1* transcript by MseI digestion. (7) RT-PCR primers used to amplify cDNA made from *bGuy1C* and *bGuy1N* transcripts. A set of RPS4 primers different from set 2 was used.

## Discussion

### Is *Guy1* the *M* factor in *An. stephensi*?

The *M* factor is a primary signal from the male-determining chromosome or locus that triggers male development. We have previously shown that *Guy1* is the earliest expressed Y chromosome gene and it is transcribed at the onset of maternal-to-zygotic transition, prior to sex-determination (*Criscione et al., 2013*). We have now shown in transgenic lines that the native *Guy1* promoter is able to direct *Guy1* transcription as well as GUY1 function when placed in the autosome of genetic females which lack the Y chromosome. Thus, *Guy1* is a primary signal, not a secondary signal, from the Y chromosome. We have also provided evidence that the small GUY1 protein is the functional molecule by analyzing various *Guy1* mutants or constructs in transgenic lines. When ectopically expressed in females, the small GUY1 protein confers 100% female lethality during embryonic and early larvae stages. As discussed in detail below, manipulations of master switches of sex-determination also cause sex-specific lethality in fruit flies and nematodes (*Thomas et al., 2012*; *Schutt and Nothiger, 2000*; *Penalva and Sánchez, 2003*). Taken together, we suggest that *Guy1* is a strong candidate for the *M* factor in *An. stephensi*. Unlike the newly discovered *M* factor *Nix* in *Ae. aegypti* (*Hall et al., 2015*), ectopic expression of *Guy1* did not produce masculinized XX females. However, a recently reported *M* factor in *An. gambiae*, gYG2/YOB, also confers female-specific lethality instead of sex conversion in a transient embryonic assay using in vitro synthesized YOB mRNA (*Krzywinska et al., 2016*). The cause of female lethality in the transgenic *An. stephensi* is unknown. A few scenarios are possible, one being related to dosage compensation, which is a mechanism known to occur in *Anopheles* mosquitoes ([*Hahn and Lanzaro, 2005*; *Mank et al., 2011*; *Jiang et al., 2015*] *Rose et al., 2016*), including *An. stephensi* (*Jiang et al., 2015*), to ensure equal levels of X-linked gene products in females (XX) and males (XY). It is intriguing that three known master switches of sex-determination, *Sex-lethal, Fem/Masc*, and *xo-lethal 1*, also regulate dosage compensation in *D. melanogaster, Bombyx mori,* and *C. elegans*, respectively ([*Thomas et al., 2012*; *Schutt and Nothiger, 2000*; *Penalva and Sánchez, 2003*; *Kiuchi et al., 2014*]). Loss of function *Sex-lethal* alleles cause female embryonic lethality in *D. melanogaster* due to mis-regulation of dosage compensation (*Cline, 1978*). Similarly, knockdown of *Masc* also casues female-specific letahlity in *B. mori*, most likely due to mis-regulation of dosage compensation (*Kiuchi et al., 2014*). If *Guy1* is also involved in the regulation of dosage compensation in *An. stephensi*, ectopic expression of *Guy1* in XX individuals may result in higher than normal levels of expression from X-linked genes, which could be lethal. Indeed, the sex-conversion phenotypes that resulted from ectopic expression of *Nix* were possible, presumably because *Ae. aegypti* does not require dosage compensation as it contains homomorphic sex-determining chromosomes (*Hall et al., 2015*).

There is no significant similarity between GUY1 and gYG2/YOB in their primary amino acid sequences. Thus, it will be difficult to determine whether Guy1 and YG2/YOB have a common evolutionary origin, especially as Y chromosome genes tend to evolve rapidly and similarities between very small proteins may not be easy to detect. However, both GUY1 and gYG2/YOB proteins are 56 amino acid long and have helical secondary structures (*Criscione et al., 2013*; *Krzywinska et al., 2016*). In addition, four out of the first five amino acid residues are identical between GUY1 and gYG2/YOB proteins. Thus, we cannot yet rule out the possibility that GUY1 and gYG2/YOB may be evolutionarily related. Regardless, it is unlikely that *Guy1* and *gYG2/YOB* are complete functional orthologs because *Guy1* is only expressed in the early embryos in *An. stephensi* (tapering off at 8–12 hr after egg laying, *Criscione et al., 2013*) while *gYG2/YOB* is expressed in the early embryos as well as in later developmental stages in *An. gambiae* (*Hall et al., 2013*; *Krzywinska et al., 2016*). Our hypothesis is that *Guy1* is the initial or primary signal which triggers a cascade of events downstream. This is analogous to the early *Sex-lethal* gene product (SXL$_e$) in *D. melanogaster* (reviewed in [*Herpin and Schartl, 2015*]), which is only produced during the early embryonic stage. This transient primary signal, SXL$_e$, is sufficient to trigger the female-specific cascade of events that are then maintained by downstream factors such as the late SXL protein throughout development. The transient expression of *Guy1* suggests that it may function as a trigger or initiating signal, but not as a gene that maintains sexual identity. If additional Y genes are indeed required for either initiating or maintaining sex determination in *An. stephensi*, ectopic expression of *Guy1* alone may not be sufficient to change *dsx* splicing in female cells/embryos due to the lack of Y chromosome. The transient and early embryonic expression of *Guy1* also made it difficult to knockdown *Guy1* simply because there

is not enough time for RNAi to work in the early embryos. However, further characterization of other Y genes in *An. stephensi* may enable the experimental demonstration of potential *Guy1* targets, which could lead to the elucidation of the *Guy1*-dependent developmental pathway in the early embryos.

## Complete penetrance, stable inheritance, early-stage lethality, and reproductive competitiveness indicate the potential for powerful applications

Our results demonstrated, for the first time in mosquitoes, that a Y chromosome gene, namely *Guy1*, confers 100% female lethality when ectopically expressed from an autosome in XX individuals and this effect can be stably inherited for many generations. Recent work showed that ectopic expression of an *M* factor *Nix* in *Ae. aegypti* initiated male development in genetic females (*Hall et al., 2015*) and embryonic injection of in vitro synthesized gYG2/YOB mRMA conferred female-specific lethality in *An. gambiae* (*Krzywinska et al., 2016*). However, the stability and penetrance of the phenotypes conferred by the *Nix* or gYG2/YOB transgenes remain to be determined as *Nix* or gYG2/YOB transgenic lines have yet to be reported (*Hall et al., 2015*; *Krzywinska et al., 2016*). Here, we have proved in principle that sustained sex ratio manipulation can be achieved using a Y chromosome gene with high penetrance. Furthermore, males that carry the *Guy1* transgene are more competitive in overall reproductive output than their non-transgenic male siblings under laboratory conditions. Future studies based on analysis of progeny of individual females mated with transgenic and wildtype males will give insight regarding the mating competitiveness of the transgenic individuals. Competitive assays performed in semi-field or field conditions are needed to determine whether ectopic expression of *Guy1* truly makes the males reproductively more competitive than wild type mosquitoes. Nonetheless, data presented in this study suggest that the *Guy1* transgene shows no obvious detrimental effects in males and may be developed as a new method to generate male-only mosquitoes for release.

As mentioned earlier, transgenic lines that produce only male progeny will improve current sex-separation methods and provide new ways to reduce mosquito population and disease transmission. The *Guy1* transgenic lines offer several attractive features in this regard. With regard to sex separation, 100% female lethality is ideal for both effectiveness and ethical considerations of not accidentally releasing females. In addition, released females can compete with wild-type females to mate with the released males simply due to their presence and even through assortative mating. Thus, the complete penetrance of the male-only phenotype, as seen in every *Guy1* transgenic line over many generations, with the exception of the negative controls, is a highly attractive feature when developing a novel genetic sex-separation approach based on *Guy1*. The current mass production of male mosquitoes relies on physical means of sex separation, which is labor-intensive, expensive, and not 100% accurate ([*Alphey, 2014*; *Gilles et al., 2014*]). Transgenic lines that express fluorescent markers in a sex-specific pattern have been developed to allow machine-based sorting of the sexes (e.g., [*Catteruccia et al., 2005*; *Marois et al., 2012*]). However, such an approach is expensive with a relatively low throughput (*Gilles et al., 2014*). Another positive feature of the *Guy1* lines is the embryonic or early larvae lethality, which negates the need to rear female larvae altogether. The *Guy1* transgenic lines also offer a few important advantages to strategies for reducing mosquito populations and disease transmission. The observed 100% female lethality will result in bias towards the non-biting males in multiple generations, which is theoretically much more efficient than classic sterile insect techniques in achieving population reduction and disease control ([*Thomas et al., 2000*; *Schliekelman et al., 2005*]). An exciting report recently showed that an artificial sex ratio distortion system based on the shredding of the *An. gambiae* X chromosome produced >95% male offspring (*Galizi et al., 2014*). The 100% female lethality described here was accomplished by simply expressing the endogenous *Guy1* gene from the autosome. Thus, the *Guy1*-based genetic strategy involves minimal genetic manipulation.

The existing *Guy1* transgenic lines are heterozygous. Because no female transgenic individuals passed the first instar, homozygous individuals cannot be produced by mating transgenic males with transgenic females. Conditional control of transgenic *Guy1* expression will enable the production of transgenic females required for generating homozygous transgenic individuals ([*Phuc et al., 2007*; *Marinotti et al., 2013*; *Fu et al., 2007*]), which is necessary for field applications. Alternatively, the *Guy1* transgene may be linked to recently developed CRISPR/cas9-based gene drives (*Gantz et al.,*

*2015*; *Hammond et al., 2016*) for rapid spread of this female-lethal gene over multiple generations. This may represent an effective yet self-limiting strategy to control mosquito-borne infectious diseases because the spread of *Guy1* could lead to severe reduction or local extinction of *An. stephensi* populations due to the lack of females (*Adelman and Tu, 2016*). Research on *Guy1* has shown great promise for genetic control of mosquito-borne infectious diseases and revealed the complex and sex-specific effect of a Y gene during mosquito embryonic development. The unexplored mosquito Y chromosome may in fact turn out to be a gold mine for intriguing stories about sexual dimorphism and promising leads for new ways to control mosquito-borne infectious diseases.

## Materials and methods

### Rearing of mosquitoes

Wild type Indian strain ('type' form) of *An. stephensi* and *transgenic An. stephensi* were reared in incubators at 27°C and 60% relative humidity on a 16 hr light/8 hr dark photoperiod. Larvae were fed Sera Micron Fry Food with brewer's yeast, and Purina Game Fish Chow. Adult mosquitoes were fed on a 10% sucrose soaked cotton pad (*Criscione et al., 2013*).

### *piggyBac* donor plasmids

All donor plasmids (*Figure 1A–D*) were constructed by inserting the gene of interest into a *piggyBac* donor plasmid backbone that contained the *piggyBac* arms and the DsRed transformation marker cassette (*Horn et al., 2000*). The *nGuy1* plasmid (*Figure 1A*) contained the native *Guy1* gene, which included the full length *Guy1* cDNA sequence (5' UTR, CDS and 3' UTR) and the previously tested *Guy1* native promoter (*Criscione et al., 2013*). The native *Guy1* gene was PCR amplified using genomic DNA from adult male *An. stephensi* as a template and primers are shown in *Table 3*. A second round of PCR was performed to adapt AscI and PacI sites to clone into the above mentioned donor plasmid backbone. The *nGuy1* plasmid and all other plasmids constructed in this study were verified by Sanger sequencing at the Virginia Bioinformatics Institute (VBI) on the Virginia Tech campus. The *Guy1m* plasmid (*Figure 1B*) is identical to *nGuy1* except for a point mutation that changed the start codon ATG to AAG. The point mutation was introduced by synthesizing and replacing the fragment between the NheI and BglII sites in *nGuy1* (Epoch Life Science, Inc., Missouri City, TX). *bGuy1C* (*Figure 1C*) contained the 168 bp *Guy1* open reading frame (ORF) plus a C-terminal Strep II tag (*22*) and the 5'UTR was provided from an *An. stephensi* early zygotic gene *bZip1* (Genbank JQ266223) and the 3' UTR from SV40. Cloning of *bGuy1C* was done by gene synthesis (Epoch Life Science, Inc., Missouri City, TX) and the sequence of the synthesized fragment is provided in the supplemental file. *bGuy1N* (*Figure 1D*) is similar to *bGuy1C* except that the Strep II tag is at the N-terminus and the 119 bp *Guy1* 3' UTR was used instead of the SV40 3' UTR. Cloning of *bGuy1N* was also done by gene synthesis (Epoch Life Science, Inc., Missouri City, TX) and the sequence of the synthesized fragment is provided in the supplemental file.

### Embryonic injection

The preblastoderm embryo injection was performed at the University of Maryland, College Park, Institute for Bioscience and Biotechnology Research's Insect Transformation Facility (http://www.ibbr.umd.edu/facilities/itf). For the *nGuy1* and *Guy1m* experiments, an injection solution was used that contains a 150 ng/μl donor plasmid, a 300 ng/μl *piggyBac* transposase (phsp-Pbac) helper plasmid (*Horn et al., 2000*), and a 200 ng/μl actin5C-EGFP plasmid. Crossing of $G_0$ and screening were performed at Virginia Tech as described below. The *bGuy1C* and *bGuy1N* lines were generated at the University of Maryland Facility using a similar injection method as the one described above. No actin5c-EGFP plasmid was included. Instead, a plasmid mixture was included for monitoring the quality of injections consisting of a minimal *Minos* vector (pMin QC) and a source of Minos transposase (pHSS6hsILMi20).

### Screening for transgenic lines

Injected embryos for *nGuy1* and *Guy1m* constructs were obtained from the University of Maryland Insect Transformation Facility and reared to adulthood. $G_0$ adult males were individually mated with five wild type females while $G_0$ females were pooled and mated with wild type males at a 1 male to

5 female ratio. Mosquitoes were blood-fed on female Hsd:ICR (CD-1) mice (Harlan Laboratories, http://www.harlan.com). Egg collection occurred approximately 3 days post blood-feeding. Larvae were screened for DsRed expression during the L3 stage. $G_1$ males and females were segregated after emergence and mated with wild type at a 1 male to 5 female ratio. The *nGuy1* and *bGuy1* lines produced no females, therefore, subsequent generations were maintained with the addition of up to 50 virgin wild type females. Females were produced in the *Guy1m* line and the line was allowed to inbreed.

## Inverse PCR and ddPCR

Inverse PCR was used to identify insertion sites of the *nGuy1* and *Guy1m* transgenes. Genomic DNA was extracted with the gDNA Isolation Kit (Zymo Research) from 10 adult males in each transgenic line: *nGuy1-1, nGuy1-2,* and *Guy1m*. Digestion of 1 μg of gDNA was performed overnight at 37°C with 80 units of restriction enzyme and 1x digestion buffer. Digestions were performed with HaeIII, PacI, and MspI and inactivated at 85°C for 5 min. The digested DNA was then allowed to self-ligate overnight at 4°C in a 200 μL ligation reaction consisting of 20 μL 10X ligation buffer, 40 μL digestion reaction, 10 μL DNA ligase, and 130 μL $H_2O$. The ligated product was then purified with the illustra GFX Gel Band Purification Kit (GE Healthcare). Purified DNA was used as the template in PCR amplification using the *piggyBac* right hand primer sets (*Table 3*) and the rTaq polymerase (Takara). Cycling conditions for the first round of PCR and nested PCR were as follows: 95°C for 3 min; 30 cycles of 95°C for 30 s, 56°C for 30 s, 72°C for 3 min; and a final extension time of 5 min. PCR products were run on a 1% agarose gel, excised and purified with the illustra GFX Gel Band Purification Kit (GE Healthcare). The purified PCR product was cloned into the pGEM-T easy vector. JM109 cells were transformed with each reaction mix and grown on LB+Ampicilin plates overnight at 37°C. Individual colonies were cultured in 3 mL LB overnight and PCR screened for the correct insertion. Plasmids were sequenced at VBI. The transgene copy number was determined by digital droplet PCR using DsRed primers and primers for an autosomal reference gene (*Table 3*) as described in Hindson *et al.* (*Hindson et al., 2011*).

## Transgene expression

Analysis of the *Guy1* transgene in the *bGuy1C* and *bGuy1N* lines were straightforward and transgene-specific primers can be used for RT-PCR (*Figure 2* and *Table 3*). For the *Guy1m* line, we were able to take advantage of the ATG to AAG mutation, which generated a MseI site that was not present in the endogenous *Guy1* transcripts. Thus, RNA was isolated from 5–6 hr embryos collected from *Guy1m* females and used for cDNA synthesis. Specific RT-PCR primers were designed to encompass a 130 bp region, which lacks the MseI site in the endogenous *Guy1* transcript. RT-PCR products were then digested with MseI to show transgene expression as indicated by the appearance of 90 and 40 bp fragments (*Figure 2*). The uncut band reflects RT-PCR products from the endogenous *Guy1* transcripts.

For *nGuy1* lines, there is no difference between the transgene and endogenous *Guy1* transcripts. In order to confirm transgene expression, we have to genotype individual embryos to show that the *Guy1* transgene is transcribed in transgenic females, which lack endogenous *Guy1*. Individual 5–6 hr old embryos were homogenized in 5 uL Lysis Buffer (0.05M DTT and 10U RNase OUT from Invitrogen). The lysate was flash frozen and stored at −80°C until gDNA preparation or RNA extraction was required. The lysate was split in half, one for genotyping and one for RT-PCR (*19*). For gDNA isolation 2.5 uL of the lysate was treated with 0.3 mAU of proteinase K for 30 min at 28°C, and then inactivated at 95°C for 2 min. The reaction was then diluted with 6.5 uL of $ddH_2O$. The gDNA of each embryo was then used as the template to amplify RPS4, *sYG2*, and the transgene in PCR reactions for genotyping purposes (see *Table 3* for primers). Transgene was amplified by primers encompassing DsRed and *Guy1*. *sYG2* is a Y chromosome gene and RPS4 is the ribosomal protein subunit 4 positive control. *sYG2* and RPS4 positive and transgene negative individuals were considered wild-type male embryos. *sYG2*, RPS4, and transgene positive individuals were considered transgenic males. Transgene and RPS4 positive and *sYG2* negative embryos were considered transgenic females. Finally, RPS4 positive and *sYG2* and transgene negative individuals were considered wild-type females. For RNA isolation 2.5 uL of the lysate was treated with 130U of DNase I (Invitrogen, Carlsbad, CA) for 1 hr at 25°C. The reaction was stopped with the addition of 1 μL of

25 mM EDTA and incubation at 65°C for 10 min. The resulting 4.5 µL reaction was carried over to cDNA synthesis using the SuperScript III reverse transcriptase and random hexamers. The cycling conditions for all reactions were as follows: 95°C for 5 min; 40 cycles at 95°C for 30 s, 60°C for 30 s, and 72°C for 3 min; and final extension at 72°C for 5 min. PCR products were run on a 1% agarose gel (*Figure 2*).

## Reproductive competitiveness of transgenic males

Mating competence of transgenic males was measured through competition against non-transgenic sibling males for mates with wild type females. F1 transgenic males were mated with wild type virgin females. The resulting F2 progeny was checked for expression of DsRed and for the presence of testes. Males that were DsRed positive and DsRed negative (siblings) were reared separately in pans with equal larval density. Two days after emerging, twenty transgenic males and twenty non-transgenic males were placed in a 44 oz Coke cup (44TDCB) together. Ten wild type virgin females (approximately three days post pupal emergence) were introduced and allowed to mate for twenty-four hours with the mixed males. The females were then blood fed for a minimum of thirty minutes and inspected for feeding. Seventy-two hrs post blood feeding, eggs were collected from these females for 24 hrs. The subsequent F3 progeny were screened for DsRed and sexed at the L3 larval stage. Biological triplicates were performed for these sets of experiments. The expected results for transgenic males, non-transgenic females, and non-transgenic males were 1/7, 3/7, and 3/7, respectively (see note of *Figure 4—source data 1* for detailed explanations). It has already been experimentally determined that transgenic females do not survive to the L3 stage and are, thus, not calculated into the expected progeny ratios.

## Acknowledgement

This work was supported by NIH grants AI105575, AI77680, and AI121284 to ZT and by the Virginia Experimental Station. We thank Jim Biedler, Janet Webster, Brantley Hall for critical review of the manuscript; Wanqi Hu for RT-PCR; Jim Biedler for cloning of some of the *Guy1* constructs; Brantley Hall and Xiaofang Jiang for revising the figures; Randy Saunders for mosquito care. We thank Robert Harrell of the Insect Transgenic Facility at the University of Maryland for embryonic injections and for generating the *bGuy1C* and *bGuyN* lines. The X-linked CFP line was provided by Anthony James of the University of California, Irvine.

## Additional information

### Funding

| Funder | Grant reference number | Author |
| --- | --- | --- |
| National Institute of Allergy and Infectious Diseases | AI105575 | Zhijian Tu |
| National Institute of Allergy and Infectious Diseases | AI121284 | Zhijian Tu |
| Virginia Experimental Station | HATCH project 135998 | Zhijian Tu |
| National Institute of Allergy and Infectious Diseases | AI77680 | Zhijian Tu |

The funders had no role in study design, data collection and interpretation, or the decision to submit the work for publication.

### Author contributions

FC, Conception and design, Acquisition of data, Analysis and interpretation of data, Drafting or revising the article; YQ, Acquisition of data, Analysis and interpretation of data, Drafting or revising the article; ZT, Conception and design, Analysis and interpretation of data, Drafting or revising the article

**Author ORCIDs**

Zhijian Tu, http://orcid.org/0000-0003-4227-3819

**Ethics**

Animal experimentation: This study was performed in strict accordance with the recommendations in the Guide for the Care and Use of Laboratory Animals of the National Institutes of Health. All of the animals were handled according to approved institutional animal care and use committee (IACUC) protocol (#16-067) of Virginia Tech.

## Additional files

**Supplementary files**

• Supplementary file 1. Number of male and female transgenics (DsRed positive) in the *nGuy1* and *Guy1m* lines.

• Supplementary file 2. Number of transgenic (DsRed positive) males in the *bGuy1C* and *bGuy1N* lines.

• Supplementary file 3. Supplemental sequences.

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
