## [Decision Letter]

Thank you for submitting your article "GUY1 confers complete female lethality and is a strong candidate for a male-determining factor in Anopheles stephensi" for consideration by *eLife*. Your article has been reviewed by two peer reviewers, and the evaluation has been overseen by a Reviewing Editor and Detlef Weigel as the Senior Editor. The following individuals involved in review of your submission have agreed to reveal their identity: Flaminia Catteruccia (Reviewer #1); Giuseppe Saccone (Reviewer #2).

Your manuscript was reviewed favorably, and the reviewers have provided straightforward and constructive suggestions. Because the reviewers are in agreement in supporting the publication of your manuscript, and the suggestions are all straightforward, we decided to simply pass their comments to you.

*Reviewer #1:*

This is a solid and interesting study that shows that a Y chromosome gene, *Guy1*, previously identified by the authors, confers female lethality in *Anopheles stephensi* mosquitoes when autosomally expressed under a germ line specific promoter. The authors elegantly confirm the specificity of *Guy1*-induced female lethality using control lines where this gene is mutated. The results clearly show that *Guy1* may be used to generate male-only populations for transgenic releases, once a system for conditional expression is developed and optimized. Moreover, they are consistent with a role for *Guy1* as a male determining factor in these mosquitoes, although they don't prove it. The authors suggest dosage compensation as a possible mechanism for the observed female lethality but they don't test this hypothesis. This study therefore is a promising first step towards the generation of a potential new tool for mosquito control and towards the elucidation of the sex determination pathway in this *Anopheles* species, although neither is fully accomplished.

The experimental procedures are well performed using a variety of controls and multiple transgenic lines. My only issue concerns the mating competition experiments described in Figure 4, which are difficult to interpret. Mating competitiveness is better measured by characterizing the progeny of each individual female rather than counting bulk numbers. How many females mated in 24 hours? And of these, how many mated to wild type versus transgenic males? What was the fecundity and fertility of the two groups? The 'wild type' males chosen in the experiments are non-transgenic *Guy1* siblings: what's the standard mating rate, fecundity and fertility of females mated to these males? In their current format I do not think those experiments unequivocally show that *Guy1* transgenics are more competitive than wild type males. If confirmed, it would be great if the authors could speculate on how *Guy1* may increase mating competitiveness, which in some of the competition experiments appears incredibly high.

*Reviewer #2:*

The authors submitted a manuscript having the title:

GUY1 confers complete female lethality and is a strong candidate for a male-determining factor in *Anopheles stephensi*.

They discovered that in the mosquitoes, *An stephensis*, female-specific lethality is unexpectedly provoked by a transgene from the Y-linked *Guy1* gene, when ectopically expressed in XX individuals using an integrated copy on an autosome.

The authors designed and performed an elegant experiment, using double transgenic fluorescence to mark one X chromosome and the autosomal *Guy1* transgene, showing that its presence in XX individuals is lethal at early larval stages.

The *Guy1* gene encodes a short protein, with no sequence homology with known proteins. So it is not clear which is its molecular function, neither it has been shown which is its normal developmental function in XY individuals. The gene is transiently expressed during early stages of XY embryogenesis, and hence it could be implicated in male-specific functions, considering that its presence is not tolerated in XX individuals. Based on previous *Drosophila* genetics studies on sex-specific lethal mutations, the most likely hypothesis for *Guy1* function would be its involvement in controlling dosage compensation of X-linked genes.

This novel data could help in future to develop an effective yet self-limiting strategy to control mosquito-borne infectious diseases because the spread of *Guy1* could lead to severe reduction or local extinction of *An. stephensis* populations due to the lack of females. Hence this study shows novel findings which can be used in the near future for consistent societal benefits consisting in protection of the human health. This study deserves publication in *eLife*, after some revisions.

Indeed, less clear is the potential role of *Guy1* in male sex determination, as stated already in the title. The authors missed to offer any experimental evidence for it. So I would suggest the authors to reformulate the title, as well as the rest of the manuscript, accordingly to the data presented.

The authors missed to offer proof that this gene is indeed promoting also maleness and male development in XY. They offered clear and convincing proof that the gene, when artificially added to autosomes as a transgene, is lethal dominant to XX individuals. A double dose of *Guy1* in XY individuals (the Y-linked copy and an autosomal copy) is compatible with maleness and, again unexpectedly, even confers higher male competitiveness for mating in lab conditions.

The authors could perform an RT-PCR on XX dying transgenic larvae, to check if the dsx splicing pattern is shifted to the male-specific type. If the authors would provide such data, they could propose GUY1 as a strong candidate for a male determining factor.

---

## [Author Response]

*Reviewer #1:*

This is a solid and interesting study that shows that a Y chromosome gene, Guy1, previously identified by the authors, confers female lethality in Anopheles stephensi mosquitoes when autosomally expressed under a germ line specific promoter.

[…]

*If confirmed, it would be great if the authors could speculate on how Guy1 may increase mating competitiveness, which in some of the competition experiments appears incredibly high.*

We appreciate the overall positive evaluation. We agree that the experiments suggested by the reviewer will give new insights regarding mating competitiveness of the *Guy1* transgenic males. However, the experiments shown in Figure 4 are designed to assess the overall reproductive competitiveness, or overall reproductive outcome of these transgenic males. In other words, the question to be addressed was: would the transgenic males be competitive in overall reproductive output compared to their non-transgenic male siblings? This question is relevant to practical applications to be developed on the basis of these transgenic lines. We chose to compare transgenic males and their non-transgenic male siblings because we wanted to test whether insertion of a *Guy1* transgene would make the males significantly less fit in reproductive output, which, if true, would represent a challenge to the *Guy1*-based method. Our results, based on three biological replicates each for two independent *Guy1* insertion lines, showed no evidence for reduced fitness under our experimental conditions.

We agree that it will be useful to understand to what extent the *Guy1* transgene may specifically affect mating competitiveness. However, we feel that more informative experiments from an application perspective are to determine the fitness and behavior of the *Guy1* transgenic males in semi-field or even controlled field conditions, with wildtype males and wildtype females from the target mosquito populations, which are beyond the scope of this current study.

We have now clarified the above points in the Discussion section.

*Reviewer #2:*

*The authors submitted a manuscript having the title:*

*GUY1 confers complete female lethality and is a strong candidate for a male-determining factor in Anopheles stephensi.*

[…]

*This novel data could help in future to develop an effective yet self-limiting strategy to control mosquito-borne infectious diseases because the spread of Guy1 could lead to severe reduction or local extinction of An. stephensis populations due to the lack of females. Hence this study shows novel findings which can be used in the near future for consistent societal benefits consisting in protection of the human health. This study deserves publication in eLife, after some revisions.*

We appreciate the overall positive evaluation of our manuscript.

*Indeed, less clear is the potential role of Guy1 in male sex determination, as stated already in the title. The authors missed to offer any experimental evidence for it. So I would suggest the authors to reformulate the title, as well as the rest of the manuscript, accordingly to the data presented.*

*The authors missed to offer proof that this gene is indeed promoting also maleness and male development in XY. They offered clear and convincing proof that the gene, when artificially added to autosomes as a transgene, is lethal dominant to XX individuals. A double dose of Guy1 in XY individuals (the Y-linked copy and an autosomal copy) is compatible with maleness and, again unexpectedly, even confers higher male competitiveness for mating in lab conditions.*

*The authors could perform an RT-PCR on XX dying transgenic larvae, to check if the dsx splicing pattern is shifted to the male-specific type. If the authors would provide such data, they could propose GUY1 as a strong candidate for a male determining factor.*

We first address the two specific comments: 1) RT-PCR to detect possible shifts in *dsx* splicing in dying XX transgenic first-instar larvae could offer proof that *Guy1* functions in male-determination. We have previously attempted to perform RT-PCR and qRT-PCR on the newly hatched wildtype first instar larvae. However, we could not detect any sex-specific *dsx* transcripts at the L1 instar stage. Therefore, it has not been technically possible to perform the experiment suggested by the reviewer, possibly due to low abundance or lack of sex-specific expression of *dsx* in most cells of the newly hatched larvae. Even if sex-specific *dsx* splicing could be detected in wildtype early first instar, there is a more fundamental challenge. Because *Guy1* is only transiently expressed in the early embryos in *Anopheles stephensi*, it cannot function as a gene that maintains sexual identity. This is different from the situation in *Anopheles gambiae*, where *gYG2/YOB* is expressed throughout development. Ectopic expression of *Guy1* alone in XX embryos or cells may not be sufficient to change *dsx* splicing if additional Y factors are required. 2) In XY individuals, because *Guy1* is expressed transiently in the early embryo (tapering off at 8hours after egg laying), it has been difficult to knock it down simply because there is not enough time for RNAi to work in the early embryo. We are developing CRISPR/cas9-based knockout in *Anopheles stephensi*. However, *Guy1* knockout may be lethal to males. To determine the mechanism of *Guy1* function, we are in the process of identifying potential *Guy1* targets, which will be informative but beyond the scope of this manuscript.

In regards to the suggestion to reformulate the title and the rest of the manuscript to better reflect the findings, we have clarified specific questions from reviewers in the text. However, we respectfully disagree with reviewer 2 that the title needs to be revised.

Here is our rationale. In the earlier summary statement, the reviewer identified key findings that are consistent with *Guy1* being a candidate for a male-determining factor, namely its Y-linkage, early embryonic expression, and its ectopic expression causing female-specific lethality (a phenomena shared with known master regulators of sex-determination in other species). We will also add two points here: 1) we have also shown, using *Guy1* transgenic lines, that *Guy1* transcription from its native promoter does not require any other factors from the Y chromosome. Thus, *Guy1* is an initial or primary signal from the Y chromosome. Please note that we are not saying that this is the proof for *Guy1* being the initiating male-determining factor. We are saying that this is evidence that *Guy1* is a primary signal, not a secondary signal from the Y chromosome. 2) GUY1 is of the same length as the recently reported M factor in *Anopheles gambiae* gYG2/YOB (56 aa) and shares similar secondary structure with gYG2/YOB. Although there is no significant primary sequence similarity between GUY1 and gYG2/YOB over the whole sequence, four out of the first five amino acid residues are identical between these two proteins. We are not at all claiming that the two proteins share a common origin. But neither can we rule out this possibility because Y chromosome genes tend to evolve rapidly and similarities between very small proteins may not be obvious. We agree with the reviewer that additional evidence is needed to conclude that *Guy1* is the male-determining factor. However, taken all existing data together, we think it is appropriate to say that *Guy1* is a strong candidate for an M factor in *Anopheles stephensi*.

We have now clarified the above two points in the first section of the Discussion.